# Patient Acceptability of Home Monitoring for Neovascular Age-Related Macular Degeneration Reactivation: A Qualitative Study

**DOI:** 10.3390/ijerph192013714

**Published:** 2022-10-21

**Authors:** Seán R. O’Connor, Charlene Treanor, Elizabeth Ward, Robin A. Wickens, Abby O’Connell, Lucy A. Culliford, Chris A. Rogers, Eleanor A. Gidman, Tunde Peto, Paul C. Knox, Benjamin J. L. Burton, Andrew J. Lotery, Sobha Sivaprasad, Barnaby C. Reeves, Ruth E. Hogg, Michael Donnelly

**Affiliations:** 1School of Psychology, Queen’s University of Belfast, Belfast BT7 1NN, UK; 2Centre for Public Health, Queen’s University of Belfast, Belfast BT12 6BA, UK; 3Bristol Trials Centre (CTEU), University of Bristol, Bristol Royal Infirmary, Bristol BS2 8HW, UK; 4Southampton Clinical Trials Unit, University of Southampton, University Road, Southampton SO17 1BJ, UK; 5Exeter Clinical Trials Unit (EXECTU), University of Exeter, St. Lukes Campus, Exeter EX1 2LT, UK; 6Department of Eye and Vision Science, University of Liverpool, Liverpool L7 8TX, UK; 7James Paget University Hospitals NHS Foundation Trust, Norfolk NR31 6LA, UK; 8Department of Clinical and Experimental Sciences, Faculty of Medicine, University of Southampton, Southampton SO16 6YD, UK; 9NIHR Moorfields Biomedical Research Centre, Moorfields Eye Hospital NHS Foundation Trust, London EC1V 2PD, UK

**Keywords:** patient perspective, technology acceptance, ophthalmic care, qualitative methods

## Abstract

Neovascular age-related macular degeneration (nAMD) is a chronic, progressive condition and the commonest cause of visual disability in older adults. This study formed part of a diagnostic test accuracy study to quantify the ability of three index home monitoring (HM) tests (one paper-based and two digital tests) to identify reactivation in nAMD. The aim of this qualitative research was to investigate patients’ or participants’ views about acceptability and explore adherence to weekly HM. Semi-structured interviews were held with 78/297 participants (26%), with close family members (n = 11) and with healthcare professionals involved in training participants in HM procedures (n = 9) (n = 98 in total). A directed thematic analytical approach was applied to the data using a deductive and inductive coding framework informed by theories of technology acceptance. Five themes emerged related to: 1. The role of HM; 2. Suitability of procedures and instruments; 3. Experience of HM; 4. Feasibility of HM in usual practice; and 5. Impediments to patient acceptability of HM. Various factors influenced acceptability including a patient’s understanding about the purpose of monitoring. While initial training and ongoing support were regarded as essential for overcoming unfamiliarity with use of digital technology, patients viewed HM as relatively straightforward and non-burdensome. There is a need for further research about how use of performance feedback, level of support and nature of tailoring might facilitate further the implementation of routinely conducted HM. Home monitoring was acceptable to patients and they recognised its potential to reduce clinic visits during non-active treatment phases. Findings have implications for implementation of digital HM in the care of older people with nAMD and other long-term conditions.

## 1. Introduction

Age-related macular degeneration (AMD) is a chronic, progressive condition and the commonest cause of vision loss in older adults [1]. Global prevalence of AMD is predicted to increase from 196 million in 2020, to 288 million in 2040 [2]. Neovascular AMD (nAMD) is a form of late AMD and is often associated with irreversible visual loss. It accounts for around 90% of cases of severe sight impairment [3]. Ongoing surveillance is necessary to manage disease activity since nAMD can recur following periods of treatment [4]. The delivery of high-quality care for nAMD relies on effective pathways that ensure maintenance of adequate review intervals to ensure active treatment is delivered appropriately, while allowing for lower risk patients, and those who would not currently benefit from treatment to be safely discharged [5,6]. Adequate referral pathways therefore need to be in place for patient reassurance, to identify suspected reactivation and for rapid referral in the case of involvement of a second eye [7,8,9,10]. Home monitoring (HM), as a form on ongoing disease surveillance, could potentially afford greater convenience for patients and family members (in terms of reducing frequency of clinic monitoring visits) and may lead to decreased costs for services, particularly regarding outpatient clinics. The COVID-19 pandemic has also highlighted the need to examine models of remotely delivered ophthalmology care, including use of HM for diagnostic purposes [11,12,13,14].

Mobile Health (mHealth) refers to use of devices including mobile phones, tablet computers or patient monitoring devices to detect and monitor changes in patient’s health and illness status [15]. This can include “passive” monitoring of behaviour, including physiological data, and capture of “active” sensor data to measure symptoms [16]. mHealth has been used to monitor chronic conditions, particularly among older people including patients with nAMD [17,18,19,20]. However, views about acceptability of HM are unclear. ‘Acceptability’ reflects the extent to which patients consider HM appropriate and feasible, based on anticipated or experienced cognitive and emotional responses [21,22]. The exploration and integration of patient preferences is important in relation to development and implementation of recommendations for diagnostic testing [23]. This study formed part of a multi-centre diagnostic test accuracy cohort study (The MONARCH Study) [24] which quantified the ability of three, non-invasive index HM tests to detect reactivation of nAMD, in comparison to a reference diagnosis of reactivation made by an ophthalmologist in a usual care nAMD monitoring clinic. The index tests were the paper-based KeepSight Journal (KSJ) [25], and two digital tests, the MyVisionTrack^®^ (mVT) [26] and MultiBit test (MBT) [27] Apps. The primary aim was to determine participants’ views about the acceptability of using the index tests. In addition, we explored adherence to weekly HM, and examined perspectives of family members and healthcare professionals providing support to participants as part of HM, including training patients for the study.

## 2. Materials and Methods

Qualitative methods were used to explore individual responses, views and experiences around HM acceptability, as well as to examine variations in contexts [28]. Semi-structured interviews were conducted face-to-face and via telephone. Participants were unknown to researchers conducting the interviews. The interview schedule (see Appendix A) was based on the experience of the research team and on theories of technology acceptance. These included the Unified Theory of Acceptance and Use of Technology (UTAUT) [29,30,31], the Technology Acceptance Model (TAM) [32,33], the Theoretical Framework of Acceptability (TFA) [34,35] and the Senior Technology Acceptance Model (STAM) [36,37]. Members of the team that collected and analysed data (CT, SOC, MD) have extensive experience in the application of qualitative methods in healthcare research. The study followed the consolidated criteria for reporting qualitative research (COREQ) criteria [38]. Ethical approval was acquired from the National Research Ethics Service (IRAS ref: 232,253 REC ref: 17/NI/0235).

### 2.1. Participants

Recruitment to the MONARCH study began in July 2018 at five sites across the United Kingdom [24]. Apps were pre-installed on an iPod touch device given to participants who were asked to complete weekly HM for a minimum of 12 months, or until study completion in September 2021. Key characteristics of the three index tests for HM of nAMD reactivation are highlighted in Table 1. Recruitment to the qualitative component began three months after the MONARCH study. During the consent process for participation in the diagnostic accuracy study, individuals who consented to further contact to discuss participation in this qualitative study were approached via a telephone call from a qualitative researcher (CT, SOC). Informed consent was obtained prior to interviews and following explanation of procedures. Maximum variation sampling was used to ensure a range of perspectives were captured relating to age (young-old 50–69 years and older-old 70+ years), gender, laterality of nAMD (unilateral and bilateral) and time since first treatment (6–17 months, 18–29 months and 30–41 months). Usage data was assessed to classify participants based on adherence to HM as: ‘Regular’ (completed weekly HM without two or more gaps in testing of greater than three weeks), or ‘Irregular’ testers (stopped and started testing on more than two occasions, or stopped testing completely). Patients who declined to participate in MONARCH but consented to be contacted about the qualitative study were also approached. We approached informal ‘carers’, supporters or significant others in the lives of patients; and healthcare professionals who interacted with participants at study sites visits, to gather their perspectives about HM acceptability.

### 2.2. Data Collection and Analysis

All interviews were audio-recorded and transcribed. A directed content analysis approach based on deductive and inductive coding was used [39]. An initial coding scheme was developed (CT, MD) thatwas based on a synthesis of relevant theoretical construct of technology acceptability [29,30,31,32,33,34,35,36,37]. see Appendix A. The coding framework underwent iterative development as individual transcripts were reviewed and re-reviewed during data familiarisation (CT, SOC, MD). Following line-by-line coding of each transcript (CT, SOC), findings were summarised based on the coding scheme and these summaries were used to revise initial codes if necessary and develop new codes based on emerging data (See Appendix A for final coding scheme). A third researcher (MD) coded a random sample of 10% of the transcripts and subsequently discussed and compared coding with CT and SOC in order to ensure adequate rigour and reflexivity. Respondent validation was also undertaken for approximately 10% of the interviews. Related codes were then clustered and grouped into initial themes. Narrative summaries were written (SOC) for each theme, then reviewed (CT, MD) and discussed to refine main themes and sub-themes to ensure coherence. NVivo version 12 was used to manage data and facilitate the analysis process, which in summary included the following stages: i. Independent transcription, ii. Data familiarisation, iii. Independent coding, iv. Development of an analytical framework, v. Indexing, vi. Charting and vii. Interpreting data.

## 3. Results

Thirty percent (89/297) of participants recruited to the MONARCH study agreed to be contacted about the qualitative component. When contacted-3/89 did not wish to take part, and 8/89 could not be reached. The remainder (26%; 78/297 of MONARCH participants) were interviewed (see Figure 1). This included participants categorised as “regular” (n = 63) or “irregular” testers (n = 14) and “non-testers” who declined to take part in HM (n = 1). Characteristics of patient participants (n = 78) were comparable to those not taking part in the qualitative study (see Table 2). In addition to the 78 patient participants, 11 informal ‘carers’, and 9 healthcare professionals were interviewed (6/11 informal ‘carer’ interviews took place in the presence of the patient participant). A total of 98 interviews were completed (patients, carers and health professionals). Interviews were conducted, in-person at patients’ homes (n = 51), at clinical sites (n = 4), or via telephone (n = 45) between October 2018 and September 2020, lasting 36 min on average (range: 25 to 78 min).

Views about HM acceptability appeared to be represented by five overarching themes (and nine associated sub-themes): 1. The role of HM; 2. Suitability of procedures and instruments; 3. Experience of HM, and 4. Feasibility of HM in usual practice; 5. Impediments to home monitoring. Relationships between themes and coding of data are shown in Appendix A. Each theme is presented in the section below. Illustrative quotes are provided in Table 3. Views of informal ‘carers’ and healthcare professionals are summarised in Appendix A.

### 3.1. Theme 1. The Role of Home Monitoring

#### 3.1.1. Sub-Theme 1: Understanding Purpose

The role of HM was clear from the perspective of participants who viewed it as facilitating ‘self-measurement’ of visual acuity. Monitoring was viewed as providing ‘ownership’ or ‘control’ over visual health, in that patients ‘knew their own eyes better than anyone else’. Participants described how they ‘self-monitored’ vision before the study, noting ‘…changes when looking out at a street sign visible from my window’, or checking for difficulties when reading or watching television. This practice is referred to clinically as an ‘environmental Amsler test’.

#### 3.1.2. Sub-Theme 2: Perceived Impact on Eye Care

Home monitoring was seen as supporting usual care by providing further assessments of possible visual deterioration. Participants did not regard HM as a ‘replacement’, but as part of their care, particularly during non-active treatment periods. It was recognised as potentially reducing the frequency of clinic visits. This was an acceptable and positive outcome. It was highlighted this could save time and effort without inconveniencing family and friends often required to transport patients to clinics. Potential for HM to reduce health service costs and relieve burden on services was noted. It was recognised this might allow appointments to be targeted based on clinical need, specifically, when risk of deterioration is higher. Concerns were expressed that reliance on HM might delay treatment depending on the degree to which results were reviewed by clinic staff.

### 3.2. Theme 2. Suitability of Procedures and Instruments

The iPod touch and index tests were viewed as novel innovations but also as reflecting the increasing pervasiveness of technology. Participants’ pre-HM use of similar devices varied but even those with minimal exposure viewed HM as realistic and suitable. Overcoming unfamiliarity or hesitancy with technology was regarded as ‘something that needed to be done’ as part of their care. Participants suggested demonstrations of the tests, prior to, and separate from the formal ‘training’ in HM procedures provided reassurance that it was ‘simple enough to do’. This increased belief in their capacity to undertake HM. In addition to technological apprehension, participants reflected that being older and unfamiliar with technology might limit engagement, with concerns that HM was too complex or ‘burdensome’.

### 3.3. Theme 3. Experience of Home Monitoring Procedures

#### 3.3.1. Sub-Theme 1: Training for Home Monitoring

Participants described training as essential for ‘getting going’ with HM but as ‘information heavy’. ‘Refresher’ sessions were suggested to help overcome difficulties recalling training information. Experiential learning was important as participants reported HM became easier and routine with practice. Participants described as valuable ‘informal’ support and advice that health professionals provided during clinic-based study visits.

#### 3.3.2. Sub-Theme 2: Test Preferences

Participants viewed the iPod as easy to use but suggested a larger screen could increase usability. Tests were referred to, positively, as ‘feeling like a game’ that reduced ‘boredom’ of repeated testing. A preference was expressed for MultiBit (MBT) relative to MyVisionTrack^®^(mVT). This was linked to ‘feedback’ (represented as a percentage score) (discussed under sub-theme 3 below). Other factors influenced participants’ views. For MBT, there was lack of clarity about the purpose of the later stages of the test, described as ‘disheartening’ as they became progressively ‘too fast’. The mVT test was viewed as too subtle (when discriminating between the ‘distorted’ circles), and as being difficult to complete, leading to participants ‘guessing’ responses. In addition, the mVT test and the paper-based keepSightJournal (KSJ) were perceived as less engaging compared to the MBT. Overall, MBT test scores were viewed as providing valuable feedback to ‘self-monitor’ changes in vision. This was apparent despite participants not being encouraged to use the scores, or being provided with information on their purpose. Participants described noting results, making comparisons over time and using feedback to ‘beat my last score’. This sense of ‘self-competition’ was seen as helpful for maintaining engagement. However, perceived ambiguity about meaning of scores produced uncertainty over how to respond to changes, and doubts about how results were used, and regarding their role as a patient. ‘Retesting’ was described as a useful way of confirming changes in scores. Negative changes were attributed sometimes to tiredness or becoming distracted during testing. Consistently lower results, even when small in magnitude, were interpreted as a concern about their eye health.

### 3.4. Theme 4. Feasibility of Regular Home Monitoring in Usual Service Delivery

#### 3.4.1. Sub-Theme 1: Frequency of Home Monitoring and Habit Formation

Participants highlighted how their views about HM improved as they became more familiar with testing. Weekly HM was seen as realistic and feasible, taking between 10–15 min. Establishing a HM ‘routine’ was considered important. Participants used several methods to help continue regular HM, including written reminders or prompts. It was described how family members would visit at set times each week and this functioned as a reminder when to test, and a means of getting help with setting up devices. Participants referred to using other forms of digital ‘self-monitoring’, including blood pressure and respiratory rate measurements; and said this made it easier to set up HM routines, acting as a reminder to complete vision tests. Participants acknowledged that HM would need to be continued in the longer-term and would need to be ‘easy, and not a burden’ to achieve sustainability and high adherence.

#### 3.4.2. Sub-Theme 2: Use of Ongoing Support

Support provided by the study telephone helpline was perceived as key. Occasionally, advice were provided by health professionals during clinic visits and this was deemed valuable.

### 3.5. Theme 5. Impediments to Home Monitoring

#### 3.5.1. Sub-Theme 1: Practical Issues

Practical or technical issues with HM were relatively infrequent. These tended to be minor, including devices needing to be recharged before testing (see Appendix A). Occasionally, issues stopped participants from testing and returning to HM-restarting even after a brief period of non-testing could be challenging. Participants reported, in general, that issues could be overcome via ‘problem solving’. Regarding the MBT, participants addressed the requirement for dark conditions during testing by changing their physical environment, such as placing a blanket over their head or testing in windowless spaces. Sometimes, adaptions were described as making the test challenging and ‘awkward’. A more substantial issue was that participants were sometimes unsure if tests had been completed successfully and data had been transmitted. It was suggested that an improvement might be to provide instantaneous confirmation of both aspects.

#### 3.5.2. Sub-Theme 2: Personal Health and Social Factors

Other issues that negatively affected HM were related to health concerns or functional limitations such as fine motor tremors, fatigue and concentration problems. Participants with caregiving responsibilities, such as providing childcare, or caring for a partner or spouse, reported difficulties finding time to test consistently.

### 3.6. Views of Informal ‘Carers’ and Healthcare Professionals–Summary

There was a high level of concurrence between perspectives of informal ‘carers’ and healthcare professionals and views of patients. Healthcare professionals suggested that increasing age might be linked to reluctance or hesitancy about HM, acknowledging that this view may be related to assumptions about older people and use of technology. Healthcare professionals agreed with patients’ views that training was essential to uptake and use of HM, and that training needed to be adapted to individual patients and their technology experience. There was a concern that HM could be resource intensive in terms of technical support needed (See Appendix A). Informal ‘carers’ or family members saw their role in HM as supportive and facilitative; and valued feedback from test performance to assess response to treatment received by their family member (See Appendix A).

## 4. Discussion

This qualitative study investigates views of patients, informal ‘carers’ and healthcare professionals about acceptability of home monitoring for nAMD reactivation. Home monitoring was acceptable to participants and key factors such as patient’s understanding of HM, and how it could be integrated into usual care appeared to influence these views. According to relatively younger patients, older peers might find HM to be a challenge–a perception also reflected in views of healthcare professionals. However, the factors that appeared to have a greater impact on positive views about HM acceptability were participants’ perceptions around the usefulness of HM for eye care, how easy it was to complete weekly HM, and their experience of undertaking HM.

Inexperience with using technology did not seem to limit or affect HM, or a participant’s intention to use it, and experience relating specifically to other forms of digital monitoring of health symptoms (e.g., blood pressure monitoring or medication reminder apps) may have been a facilitating factor. Establishing the ‘habit’ of HM and integrating it into a participant’s routine seemed to be important in terms of ensuring regular use. Weekly HM was feasible though more frequent monitoring (e.g., daily testing) may be too burdensome and, therefore, less acceptable. In general, the HM tests were reported to be easy to undertake and non-burdensome. The time commitment required to undertake HM was also acceptable and, although technical difficulties were relatively infrequent, access to ongoing support was regarded as essential to successful HM, and for overcoming any unfamiliarity with use of technology. Support included ‘formal’ training and assistance from healthcare professionals with technical aspects of HM, and ‘informal’ support primarily from partners and family members in the form of encouragement and facilitation of HM, and to help manage in situ any technical issues.

It was recognised that there was potential for HM to reduce the frequency of clinic visits, particularly during non-active treatment phases. This positively viewed aspect was contingent upon the expectation and availability of an easily accessible route back to usual service arrangements if further treatment was required. The use of test performance feedback was perceived by participants as a way to ‘self-monitor’ vision, even though ‘feedback’ was provided by only one of the tests (the MBT). This was despite information having been given at the outset to participants that the HM test information would not influence their care, and that they should not contact the hospital on the basis of test results, but should do so if they noticed a deterioration in their vision.

A slight preference for the MBT test was evident, mainly due to its perceived ‘feedback’ feature, and this was reported despite the test needing to be completed in darkness, which presented an inconvenience and a practical impediment to testing. In addition to the lack of performance feedback, the mVT was also perceived as being ‘too difficult’ to complete effectively. This may have been due to the staircase procedure used to determine threshold, hence there would have been trials in which the patients struggled to see an odd one out as it honed in on the final threshold. The mVT, along with the paper-based KeepSight Journal, were also reported as less ‘engaging’ than the MBT. Performance feedback alongside motivational or supportive messages can be key components of digital interventions; and they tend to be associated with higher intervention usage and acceptance [40]. Various forms of ‘meaningful’ feedback about a patient’s health status can provide a sense of support, guidance and information, as well as giving reassurance to patients [41]. However, feedback by itself may also be an intrinsic motivator to sustained HM beyond any ‘clinically meaningful’ relevance of change in performance scores [42,43,44].

While test complexity and technical or reliability problems are common barriers to technology acceptance [45,46,47], these issues were relatively infrequent in the present study. Persistence and ‘problem solving’, as well as external support were needed to overcome them in the relatively few occasions when they did occur. Uncertainty about whether or not the digital index tests had been completed successfully and if test data had been transmitted to the study team appeared to influence views about acceptabilityParticipants suggested strongly that confirmation of successful test completion and data transmission should be provided to HM patients. As one of the participants noted, ‘…it (HM) isn’t worth it if you don’t know if the test is working properly or if no one knows what is happening on the other end’. In summary, HM was acceptable to participants. Only five of 78 participants in the qualitative sample (approximately 6%) discontinued HM. Facilitators to regular HM appeared to revolve around concerns about eye health, perceptions that HM provided a sense of control through test feedback, perceived ease of test completion and the ability to form a routine or HM habit.

### 4.1. Comparisons with Previous Literature

Our finding that HM was acceptable to patients and could be used to assess the stability of nAMD and monitor associated changes in vision is in keeping with other related research about the monitoring of visual function using digital testing [48,49,50,51]. For example, a pilot study of a tablet-based programme for detecting progression in patients with intermediate AMD indicated good agreement between HM and clinic-based testing [49]. A recent, prospective cohort study concluded that HM, using the Alleye App, was a useful adjunct to usual eye care, with high specificity and predictive value for identification of disease progression (93.8% (95% CI: 86.2–98.0%) and 80.0% (95% CI: 59.3–93.2%), respectively [50].

Despite information security having been highlighted as an issue with the use of home-based digital technology in ophthalmology care [52], participants in the current study did not express this as a concern. An important observation was that MBT scores were ‘self-monitored’ by patients and to an extent, informal ‘carers’ or relatives. These participants described noticing a ‘pattern’ in scores which they perceived as indicating a decline during the time leading up to clinic appointments to receive anti-VEGF injections and which improved in the weeks following injection. Similar observations were also reported in an earlier study where patients with nAMD using the MBT undertook more frequent testing than was suggested, and independently recorded their test scores [53].

Adherence with HM in some studies appears to be associated different factors including age and treatment status. For example, ophthalmology service patients including patients with nAMD who took up the offer of self-monitoring using the Alleye App [54] tended to be significantly younger, male, living with family and not currently in active treatment. However, within this group, actual compliance with HM was higher in those who were older, and in active treatment. It should be taken into account when making comparisons with previous literature that follow-up periods in these papers have tended to be relatively short (less than six months), meaning that the long-term impact of factors influencing HM are not certain.

### 4.2. Strengths and Limitations

Strengths of the study include the use of maximum variation sampling to recruit participants based on a range of important characteristics including gender and time since initiation of treatment. Recruitment took place across different clinical sites and geographical locations, and was successful in enrolling participants who were reflective of the overall sample from the diagnostic accuracy study. Participants’ views about acceptability were similar across these locations. The study incorporated the views of informal and professional carers and, generally, there was a high level of concurrence across these perspectives. A potential limitation of the study is that a mixture of telephone and face-to-face interviews were used. This was to allow flexibility for participants, but was also a result of the COVID-19 pandemic and the required social distancing measures that precluded home visits. Despite this, there did not appear to be any substantial variances between interviews conducted using these different methods. Although an attempt was made to recruit more participants who stopped HM, or who completed it irregularly, the majority of the sample included participants who showed good levels of adherence to weekly HM. This might suggest the sample could be described as having characteristics of ‘early adopters’ of technology, and this may have influenced the results.

## 5. Conclusions

This qualitative study provides important insights into the perspectives of patients, ‘informal’ carers and healthcare professionals about the acceptability of HM for assessing reactivation in nAMD. Home monitoring was acceptable and non-burdensome but initial training and ongoing support are essential to successful implementation. Reasons for stopping testing related to unresolved technical issues or changes in personal circumstances such as a partner becoming unwell. Patients recognised the potential for HM to reduce clinic visits, particularly during non-active treatment phases and the related cost savings. Feedback scores from index tests (MBT) were valued by participants as a way to ‘self-monitor’ visual acuity. However, there was ambiguity about use of performance scores to assess any changes in vision, and uncertainty about what scores meant in terms of visual health. These findings have important implications for the design and use of digital HM in the care of older people with nAMD as well as in other long-term health conditions.

## Figures and Tables

**Figure 1 ijerph-19-13714-f001:**
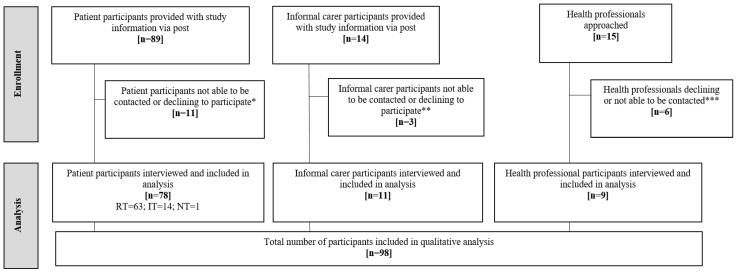
Participant flow-qualitative component of the MONARCH Study. RT = regular testers (completed weekly testing without significant gaps in testing during the study period); IT = irregular testers (completed one, two or three tests, or stopped-and-started testing, or withdrew from diagnostic accuracy study); NT = non-testers (declined to take part in home monitoring); * Declined to participate (n = 3), not contactable by telephone (n = 8); ** Not contactable by phone (n = 3); *** Moved post (n = 1), not contactable (n = 5).

**Table 1 ijerph-19-13714-t001:** Characteristics of index tests used in the MONARCH Study: the paper-based KeepSight journal (KSJ), the MyVisionTrack^®^ and the MultiBit App.

Index Test	Developer	Test Characteristics	Requested Test Frequency
KeepSight journal (KSJ)	International Macular and Retinal Foundation (New Gloucester, Maine, USA)	Paper-based format with three tests viewed one eye at a time. i. Near visual acuity test formatted as a puzzle with varying font sizes ii. A test to assess distortions by viewing objects with straight lines. iii. A modified Amsler chart to record areas of distortion or scotoma.	Weekly
MyVisionTrack^®^ (mVT)	Genentech Inc.	Shape discrimination threshold test displayed on an iPod Touch. It displays four circles, one of which is deformed. The participant identifies the odd-one-out. Participants select the odd circle out, i.e., irregularly shaped circle.	Weekly
MultiBit test (MBT)	Visumetrics, licensed by Novartis International AG	Near acuity threshold test displayed on an iPod Touch. Numbers made up of receptive field size dots or ‘rarebits’ are displayed in pairs. Participants are required to state aloud the numbers they can see. The numbers are then presented in high contrast with a recording of the participant’s responses and the participant marks their performance. The test is performed in darkness to ensure good visibility of the high contrast numbers.	Weekly

**Table 2 ijerph-19-13714-t002:** Sociodemographic characteristics of participants in the qualitative component of the MONARCH study compared to the rest of the MONARCH study sample.

		Qualitative Sample (n = 78 *)		Remaining MONARCH Study Participants (n = 221)	
		n	%	n	%
Baseline characteristics					
Sex	Male	30	38.5	93	42.1
	Female	48	61.5	128	57.9
Age	Mean (SD) years	74.3 (6.8)	-	75.1 (6.6)	-
Visual acuity **	Mean (SD) LogMAR	0.2 (0.2)	-	0.2 (0.2)	-
Smoking history	Current smoker	7	9.1	23	10.4
	Ex-smoker (>1 month)	44	57.1	94	42.5
	Never smoked	26	33.8	104	47.1
Exposure to technology					
	Television	75	97.4	220	100.0
	Simple mobile phone	24	31.2	106	48.2
	Smartphone	53	68.8	145	65.9
	Tablet	55	71.4	142	64.5
	Laptop/Home Computer	53	68.8	132	60.0
	Internet at Home	68	88.3	185	84.1
	E-mail	62	80.5	152	69.1
	Social Media	30	39.0	68	30.9
	TV streaming/On-demand services	36	46.8	110	50.0

* Calculations are based on n = 77 as overall qualitative sample includes n = 1 participant who declined to take part in home monitoring but consented to take part in the qualitative part of the study. ** For patients with two involved eyes, better seeing eye is used.

**Table 3 ijerph-19-13714-t003:** Perspectives of patients on acceptability of home monitoring.

Perspectives of Patients	Theme/Sub-Theme	Supporting Quote(s) from Patients
- HM viewed as providing ‘ownership’ or ‘personal control’ - HM could reduce the frequency of clinic visits - Clear pathways to routine clinic appointments are needed if there are changes in visual acuity	Theme 1. The role of home monitoring Sub-theme 1: Understanding purpose Sub-theme 2: Perceived impact on eye care	*‘…it is to put you in charge. I could judge if I needed help, if I saw deterioration in my vision when I did the test, or if I noticed a change by myself’.* (Female, Regular HM, 62 years, #53) *‘I would feel, yes, I’m doing the tests and that’s okay. At the minute, I’m only going (to the clinic) four times a year, so even two or three times would be okay. I’d be happy enough now [To home monitor], you know? … Providing nothing happens’.* (Female, Regular HM, 78 years, #37) *‘…I don’t think it would always work because it’s near impossible to get an appointment, you know? I mean, I’ve done that. I’ve seen a change in shape, not when I was in this study but before. I asked for an appointment but didn’t get it, so is the purpose is to try and put people more in charge of saying what they can see, saying if they need help or not?’* (Male, Regular HM, 82 years, #24)
- Overcoming unfamiliarity with technology regarded as ‘something needing to be done’ - Unfamiliarity with technology might result in hesitation about engaging in HM	Theme 2. Suitability of procedures and instruments	*‘…technology is a funny thing to lots of people my age, some have embraced it, now of course it’s a necessary evil, so I’m on catch up’* (Male, Regular HM, 76 years, #08) *‘…if this (the test device) was just given to me, I would be a bit lost but I’m always trying to keep an open mind with technology and do what I can, you know.’* (Male, Irregular HM, 79 years, #38) *‘…I mean it’s no problem because I’m not too bad. I’ve got an iPad and an iPod, but I can see lots of people couldn’t do it. A lot of them don’t even like using the computer do they?’* (Female, Regular HM, 81 years, #68) *‘…Well, mostly it’s the elderly people that have got it (AMD) and most of them are not okay with computers and things. I mean I’m not brilliant, but I can do it. As you get older you can’t learn these things so easily’.* (Female, Regular HM, 79 years, #82)
- Refresher training could help overcome difficulties recalling information - mVT and paper-based KSJ tests were perceived as less engaging than the MBT - MBT Test feedback seen as helpful for keeping engaged with HM - Lower test scores, even when small, were interpreted as a concern about their eye health	Theme 3. Experience of home monitoring procedures Sub-theme 1: Training for home monitoring Sub-theme 2: Test preferences Sub-theme 3: Use of MBT feedback and data	*‘…and so (the clinic staff) demonstrated it… I thought that actually looks easy, but a week later when I’m on my own, I just said “what did they say?’* (Female, Regular HM, 71 years, #49) *‘…well, I found that test (MBT)… first of all it was very quick. You had to be so alert and I could be pressing away and it was doing nothing because it was too fast for me’.* (Female, Regular HM, 76 years, #17) *‘…but the test with the flashing numbers (MBT), I actually liked that. I couldn’t stand the other test (mVT) because you get four shapes and one of them is sort of out of sync. The first three are easy, then it gets more and more tricky. It gets to the stage where I just had me guess. I actually found that annoying because I didn’t know how I was doing. The other one you get a percentage, which is good’.* (Male, Regular HM, 80 years, #46) *‘…so you see benefits instantly because you’ve got a result, not only have I done an exam, I have a result instantly, the minute you finish and put your stuff away, the mental benefits are there.* (Male, Regular HM, 75 years, #87) *‘…**if I get less than 90(%) then I absolutely know that there’s something wrong. I’m not happy with 92, it’s always been 94 or 96, 98, or 100. So that did worry me, but I will do it again, just to check, and I’ve got an appointment on the second anyway’.* (Male, Irregular HM, 77 years, #50)
- Several methods used to help continue regular HM, including use of reminders or prompts - Using other forms of digital ‘self-monitoring’, including blood pressure measurements; made it easier to set up a HM routine - HM needs to be ‘easy, and not a burden’ to achieve sustainability and high adherence - Family members were a source of support	Theme 4. Feasibility of regular home monitoring in usual service delivery Sub-theme 1: Frequency of home monitoring and habit formation Sub-theme 2: Use of ongoing support	*‘…and (my granddaughter) would get it set up for me and then when that test is finished, switch over on to the next but she doesn’t have to stand over me, you know.’* (Male, Irregular HM, 79 years, #38) *‘… I have used that (monitoring device for tracking COPD symptoms) for about 18 months, so this can also helped me know when I’m getting bad, because they were reading it and then they were ringing back and checking with me. That made me feel better, being in touch with people’.* (Female, Regular HM, 62 years, #53) *‘… when I first went back to [eye hospital] they gave me the bag and then when I went to [hospital] they gave me a blood pressure monitor, so what I do is, I have to check my blood pressure regularly you see, so I stick this in with my machine because I’m doing them both weekly at the minute and it all works out well, I don’t forget’.* (Female, Regular HM, 74 years, #34) *‘… my son has got me using smart phones and what not. I am ok with an iPad and an iPhone, no problem. I can handle anything in medical terms, I am keeping tabs on my medications on a daily basis. I have a little app that reminds me every hour, every two hours, what I have to do for the day’* (Male, Regular HM, 70 years, #136) *‘..You don’t do for enjoyment you’re doing it to see how it goes. I don’t look at it as a pleasure that I can’t wait to do, and think, oh I must go up and do my wobbly circles. I just think it’s time I did those, I’ll go up and do them now’.* (Female, Regular HM, 66 years, #62) *‘…I had a lot of trouble at one point, but my husband said, “let me have it,’ and he diddled about with the buttons, one of which was the light intensity so I had probably turned the light down without realising it. He helped a lot. He said ‘you go through it and see what you get stuck on. He didn’t just take over, he just said call me when you need me’.* (Female, Regular HM, 72 years, #58) *‘…so I had to ring [the helpline], he was very nice and went through it all. My son lives down the road and is into computers and I said well, I could ask my son again, but it was all sorted before my son appeared’.* (Female, Regular HM, 76 years, #16)
- Some adaptions made HM challenging and ‘awkward’. - Other health concerns or functional limitations made it harder to undertake HM - Caregiving responsibilities made it difficult to find time for regular HM	Theme 5. Impediments to home monitoring Sub-theme 1: Practical issues Sub-theme 2: Personal health and social factors	*‘…it was difficult, I just couldn’t get it dark enough. I racked my brain and thought I’ve got a big wool rug. I got under that and did my best but there’s also the claustrophobia, it just got me annoyed in the end’.* (Female, Irregular HM, 77 years, #33) *‘..and I have a tremor, when I’m holding it (the iPod), you don’t know where the numbers are going to come from on the screen… so you’re sort of anticipating you know? And this means you just don’t catch it’.* (Female, Regular HM, 71 years, #71) *‘…I have had problems with my health, my heart scare, lots of things all happening, a lot of times I think this leaves me feeling really really tired... I’m staring, not knowing if I even hit the buttons’.* (Male, Irregular HM, 74 years, #29) *‘…it’s because I have been caring for (a relative) and I don’t even remember. It’s not high on my list of priorities. I have been doing it, but it’s when I get to it, not when it gets to me’.* (Female, Regular HM, 72 years, #83)

HM: Home monitoring; mVT: MyVisionTrack^®^; KSJ: keepSightJournal; MBT: MultiBit test.

## Data Availability

Study data are available from the corresponding authors upon request.

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
