# Peer review of "Patient Acceptability of Home Monitoring for Neovascular Age-Related Macular Degeneration Reactivation: A Qualitative Study"

_ijerph, 2022, doi:10.3390/ijerph192013714_

Round 1
Reviewer 1 Report
1. The significance of the study is very necessary: in the context of the new coronary pneumonia pandemic, it is necessary to construct a model for examining remote eye care and use home monitoring for diagnostic purposes
2. The views on the acceptability of home monitoring are divided into five overarching themes and nine related sub-themes, and the content regulations for the acceptability of home monitoring are clear. Figure 4 is clear, and the summary is relatively comprehensive.
â‘ The role of home monitoring;
â‘¡ Suitability of procedures and instruments;
â‘¢ Experience of home monitoring
â‘£ Feasibility of home monitoring in usual practice
⑤ Impediments to home monitoring.
There are still certain points that the authors should improve:
1.Qualitative methods are used as key words in this paper, but the only methodological explanation of qualitative methods is a quoted quote on page 2 "Qualitative methods were used to explore individual responses, personal responses, perspectives, and experiences around the acceptability of home monitoring, and to examine changes in the environment [18]"
â‘ It is not clear what exactly is a qualitative method
â‘¡ For what reason the data in this article considered the use of qualitative methods rather than quantitative methods for research and exploration
â‘¢ Compared to quantitative studies, qualitative studies are not quantified, so statistical significance is difficult to judge whether there is a serious impairment in the strength of the evidence level, but this is not mentioned in the limitations of the article.
2.Because of the subjective nature of the interview transcripts, interview quality control for this study should be added to the methodology.
3.Table 2 presents the baseline quantitative percentage statistics, but whether the quantitative analysis was considered comparable between the two groups, the main subject of this paper was the qualitative sample (N=78), whose listed remaining MONARCH study participants (n=221) were not used, and this comparison appears in this table or in this paper, the significance is not too obvious.
4.In Table 3, the entries of â‘ Perspectives of patients â‘¡Theme/sub-theme should be switched, because the main idea of the authors of this paper was to discuss the participants' perceptions of the acceptability of using home monitoring around 5 overarching themes and 9 sub-themes.
5.In the description of the study results in the text RESULT, the author should consider counting the number of people with similar opinions.
Reviewer 2 Report
A well constructed and described piece of qualitative research which will inform the future potential of home monitoring
Author Response
Thank you for your kind comments and peer review
Reviewer 3 Report
The purpose of this study is very attractive. However, the results are too long and conclusion does not seem to written based on the current results. I have read this article for a couple of times. But I still do not recognize what findings are found in this study. Please try to summarize more clearly and shortly than what it is.
Author Response
|
ijerph-1916263 |
|
|
Acceptability of home monitoring for neovascular age-related macular degeneration reactivation: a qualitative study |
|
|
Reviewer comments |
Authors response |
|
Reviewer #3 |
|
|
1. The purpose of this study is very attractive. However, the results are too long and conclusion does not seem to written based on the current results. I have read this article for a couple of times. But I still do not recognize what findings are found in this study. Please try to summarize more clearly and shortly than what it is. |
Thank you for your comments. We have made a number of changes to reduce the overall length of the paper and specifically, the content and information included in the results section.
To summarise the findings clearly, we have removed, or moved to the supplementary files, a number of Tables and Figures in the results. We have also removed and modified paragraphs in the discussion, including the section on Technology acceptance theories, the sections on comparison with other literature, and the strengths and limitations section.
We hope the changes made give a clearer summary of the main findings which are:
· Home monitoring was seen as acceptable by participants in the study, and as straightforward and non-burdensome. · Home monitoring was also seen as a possible way to reduce clinic visits during non-active treatment phases. · Previous experience with technology was not essential for people to take part in home monitoring but training and support (from health professionals and family members) was important to overcome unfamiliarity with using digital technology. · Home monitoring provided a sense of control of a patient’s visual health, partly due to the performance feedback given by one of the tests. · Barriers to home monitoring were: · The need for a dark environment to complete one of the tests · Boredom with repeated testing. · Uncertainties around data being transferred. · Having personal health issues or caregiving responsibilities.
|
Reviewer 4 Report
Thank you for the opportunity to review this manuscript. The paper presents a detailed explanation of a home based monitoring intervention for neovascular age-related macular degeneration reactivation. It was a substudy to a larger parent study, and reports a qualitative feasibility/acceptability study.
The paper is well written, clear, comprehensive and interesting. It explains the phenomenon for unitiated readers, guiding them through the intro, methods, results and discussion clearly. There is appropriately small use of jargon or acronyms to aid understanding. The tables and figures support the methodological rigour. I have only minor comments or suggestions for consideration.
1. The paper is quite long. Should the word count need reducing to suit the journal style I suggest removing some explanatory content in the discussion and halve the length of the conclusion. If this length is appropriate, please disregard this suggestion.
2. Figure 2 caption should appear with figure 2, and should be explained in more detail in text. Currently the reader is unsure what 'distribution function estimate' means and the units, in addition to censor ticks. Please elaborate in text.
3. Figure 4 appears to be a table, rather than a figure. Please review. Additionally, it does not appear to add much for the reader and the authors could consider removing figure 4 completely.
4. Last paragraph in the first section of the discussion please clarify for the readers whether 5/78 participants discontinued home monitoring from the qual sample or larger parent study sample.
5. Consider removing references from the conclusion and only providing a succinct summary/implications.
6. In the reference list there is duplication of reference numbers. I assume this is a copy editing element which will be addressed in the next version.
Thank you for providing an interesting and important paper for me to read and review.
Author Response
|
ijerph-1916263 |
|
|
Acceptability of home monitoring for neovascular age-related macular degeneration reactivation: a qualitative study |
|
|
Reviewer comments |
Authors response |
|
Reviewer #4 |
|
|
|
Thank you for your comments. We have reduced the length of the paper including much of the explanatory content in the discussion, along with other changes to make the conclusion more succinct.
Specifically, we have removed a number of paragraphs in the discussion, including the section on Technology acceptance theories, and in the sections on comparison with other literature, and the strengths and limitation section.
|
|
2. Figure 2 caption should appear with figure 2, and should be explained in more detail in text. Currently the reader is unsure what 'distribution function estimate' means and the units, in addition to censor ticks. Please elaborate in text. |
Thank you. We agree this needs a fuller explanation and have removed the figure, as it does not contribute to the main purpose of the manuscript (as the figure relates to usage, not views on acceptability).
|
|
3. Figure 4 appears to be a table, rather than a figure. Please review. Additionally, it does not appear to add much for the reader and the authors could consider removing figure 4 completely. |
Thank you. We have moved this table to the supplementary files. |
|
4. Last paragraph in the first section of the discussion please clarify for the readers whether 5/78 participants discontinued home monitoring from the qual sample or larger parent study sample. |
Thank you for this comment. We have changed this line to clarify that it relates to the qualitative sample only.
Text amended as follows: In summary, HM was acceptable to participants. Only five of 78 participants in the qualitative sample (approximately 6%) discontinued HM.
|
|
5. Consider removing references from the conclusion and only providing succinct summary/implications. |
Thank you. References have been removed from this section. |
|
6. In the reference list there is duplication of reference numbers. I assume this is a copy editing element which will be addressed in the next version. |
Thank you for this comment. References have been checked and re-ordered. |
Round 2
Reviewer 3 Report
In the results, there are some paragraphs still missing supporting data. A result section usually contains the expression "see Figure xx" , " as shown in Figure xx or (Figure xx). Each paragraphs must express where readers should see to confirm the contents of the results. That way, it become more easier for us to figure out the new findings from this work.
Author Response
Thank you again for your comments. In this qualitative study, results/findings, and the overall conclusions that are drawn, are based on the views of participants that are elicited during the interviews. As is typical in papers that report on this type of investigation, we provide examples of participant quotes that reflect the key findings. Apologies if we have misunderstood the comment:
..paragraphs still missing supporting data. A result section usually contains the expression "see Figure xx" , " as shown in Figure xx or (Figure xx).
However, qualitative studies do not generate data that can be summarised in Figures, as would be the case for a quantitative study.
The following paragraph (Page 5) is included to highlight were illustrative quotes are provided to support the findings that are reported in the results section:
Views about HM acceptability appeared to be represented by five overarching themes (and nine associated sub-themes): 1. The role of HM; 2. Suitability of procedures and instruments; 3. Experience of HM, and 4. Feasibility of HM in usual practice; 5. Impediments to home monitoring. Relationships between themes and coding of data are shown in Supplementary file 4. Each theme is presented in the section below. Illustrative quotes are provided in Table 3. Views of informal ‘carers’ and healthcare professionals are summarised in Supplementary file 5 and 6.
In the results, there are some paragraphs still missing supporting data. A result section usually contains the expression "see Figure xx" , " as shown in Figure xx or (Figure xx). Each paragraphs must express where readers should see to confirm the contents of the results. That way, it become more easier for us to figure out the new findings from this work.